# Learning the Difference that Makes a Difference with Counterfactually-Augmented Data

**Divyansh Kaushik, Eduard Hovy, Zachary C. Lipton**
Carnegie Mellon University
Pittsburgh PA, USA
{dkaushik, hovy, zlipton}@cmu.edu

## Abstract

Despite alarm over the reliance of machine learning systems on so-called *spurious* patterns, the term lacks coherent meaning in standard statistical frameworks. However, the language of causality offers clarity: spurious associations are due to confounding (e.g., a common cause), but not direct or indirect causal effects. In this paper, we focus on natural language processing, introducing methods and resources for training models less sensitive to spurious patterns. Given documents and their initial labels, we task humans with revising each document so that it (i) accords with a counterfactual target label; (ii) retains internal coherence; and (iii) avoids unnecessary changes. Interestingly, on *sentiment analysis* and *natural language inference* tasks, classifiers trained on original data fail on their counterfactually-revised counterparts and vice versa. Classifiers trained on combined datasets perform remarkably well, just shy of those specialized to either domain. While classifiers trained on either original or manipulated data alone are sensitive to spurious features (e.g., mentions of *genre*), models trained on the combined data are less sensitive to this signal. Both datasets are publicly available[1].

## 1 Introduction

*What makes a document's sentiment positive? What makes a loan applicant creditworthy? What makes a job candidate qualified? When does a photograph truly depict a dolphin? Moreover, what does it mean for a feature to be relevant to such a determination?*

Statistical learning offers one framework for approaching these questions. First, we swap out the semantic question for a more readily answerable associative question. For example, instead of asking *what conveys a document's sentiment*, we recast the question as *which documents are likely to be labeled as positive (or negative)?* Then, in this associative framing, we interpret as *relevant*, those features that are most *predictive* of the label. However, despite the rapid adoption and undeniable commercial success of associative learning, this framing seems unsatisfying.

Alongside deep learning's predictive wins, critical questions have piled up concerning *spurious patterns*, *artifacts*, *robustness*, and *discrimination*, that the purely associative perspective appears ill-equipped to answer. For example, in computer vision, researchers have found that deep neural networks rely on surface-level texture (Jo & Bengio, 2017; Geirhos et al., 2018) or clues in the image's background to recognize foreground objects even when that seems both unnecessary and somehow wrong: *the beach is not what makes a seagull a seagull*. And yet, researchers struggle to articulate precisely why models *should not* rely on such patterns.

In natural language processing (NLP), these issues have emerged as central concerns in the literature on *annotation artifacts* and *societal biases*. Across myriad tasks, researchers have demonstrated that models tend to rely on *spurious* associations (Poliak et al., 2018; Gururangan et al., 2018; Kaushik & Lipton, 2018; Kiritchenko & Mohammad, 2018). Notably, some models for question-answering tasks may not actually be sensitive to the choice of the question (Kaushik & Lipton, 2018), while in *Natural Language Inference* (NLI), classifiers trained on *hypotheses* only (vs hypotheses and premises) perform surprisingly well (Poliak et al., 2018; Gururangan et al., 2018). However, papers

---

[1]https://github.com/dkaushik96/counterfactually-augmented-data

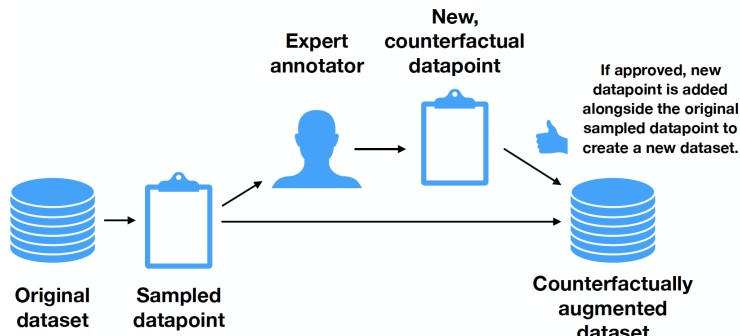

Figure 1: Pipeline for collecting and leveraging counterfactually-altered data

seldom make clear what, if anything, *spuriousness* means within the standard supervised learning framework. ML systems are trained to exploit the mutual information between features and a label to make accurate predictions. The standard statistical learning toolkit does not offer a conceptual distinction between spurious and non-spurious associations.

Causality, however, offers a coherent notion of spuriousness. Spurious associations owe to confounding rather than to a (direct or indirect) causal path. We might consider a factor of variation to be spuriously correlated with a label of interest if intervening upon it would not impact the applicability of the label or vice versa. While our paper does not call upon the mathematical machinery of causality, we draw inspiration from the underlying philosophy to design a new dataset creation procedure in which humans *counterfactually revise* documents.

Returning to NLP, although we lack automated tools for mapping between raw text and disentangled factors, we nevertheless describe documents in terms of these abstract representations. Moreover, it seems natural to speak of manipulating these factors directly (Hovy, 1987). Consider, for example, the following interventions: (i) *Revise the letter to make it more positive*; (ii) *Edit the second sentence so that it appears to contradict the first*. These edits might be thought of as intervening on only those aspects of the text that are necessary to make the counterfactual label applicable.

In this exploratory paper, we design a human-in-the-loop system for counterfactually manipulating documents. Our hope is that by intervening only upon the factor of interest, we might disentangle the spurious and non-spurious associations, yielding classifiers that hold up better when spurious associations do not transport out of domain. We employ crowd workers not *to label* documents, but rather *to edit* them, manipulating the text to make a targeted (counterfactual) class applicable. For sentiment analysis, we direct the worker to *revise this negative movie review to make it positive, without making any gratuitous changes*. We might regard the second part of this directive as a least action principle, ensuring that we perturb only those spans necessary to alter the applicability of the label. For NLI, a 3-class classification task (*entailment, contradiction, neutral*), we ask the workers to modify the premise while keeping the hypothesis intact, and vice versa, collecting edits corresponding to each of the (two) counterfactual classes. Using this platform, we collect thousands of counterfactually-manipulated examples for both sentiment analysis and NLI, extending the IMDb (Maas et al., 2011) and SNLI (Bowman et al., 2015) datasets, respectively. The result is two new datasets (each an extension of a standard resource) that enable us to both probe fundamental properties of language and train classifiers less reliant on spurious signal.

We show that classifiers trained on original IMDb reviews fail on counterfactually-revised data and vice versa. We further show that spurious correlations in these datasets are even picked up by linear models. However, augmenting the revised examples breaks up these correlations (e.g., genre ceases to be predictive of sentiment). For a Bidirectional LSTM (Graves & Schmidhuber, 2005) trained on IMDb reviews, classification accuracy goes down from $79.3\%$ to $55.7\%$ when evaluated on original vs revised reviews. The same classifier trained on revised reviews achieves an accuracy of $89.1\%$ on revised reviews compared to $62.5\%$ on their original counterparts. These numbers go to $81.7\%$ and $92.0\%$ on original and revised data, respectively, when the classifier is retrained on the combined dataset. Similar patterns are observed for linear classifiers. We discovered that BERT (Devlin et al., 2019) is more resilient to such drops in performance on sentiment analysis.

Additionally, SNLI models appear to rely on spurious associations as identified by Gururangan et al. (2018). Our experiments show that when fine-tuned on original SNLI sentence pairs, BERT fails on pairs with revised premise and vice versa, suffering more than a 30 point drop in accuracy. Fine-tuned on the combined set, BERT's performance improves significantly across all datasets. Similarly, a Bi-LSTM trained on (original) hypotheses alone can accurately classify 69% of pairs correctly but performs worse than the blind classifier when evaluated on the revised dataset. When trained on hypotheses only from the combined dataset, its performance is not appreciably better than random guessing.

## 2 RELATED WORK

Several papers demonstrate cases where NLP systems appear not to learn what humans consider to be *the difference that makes the difference*. For example, otherwise state-of-the-art models have been shown to be vulnerable to synthetic transformations such as distractor phrases (Jia & Liang, 2017; Wallace et al., 2019), to misclassify paraphrased task (Iyyer et al., 2018; Pfeiffer et al., 2019) and to fail on template-based modifications (Ribeiro et al., 2018). Glockner et al. (2018) demonstrate that simply replacing words by synonyms or hypernyms, which should not alter the applicable label, nevertheless breaks ML-based NLI systems. Gururangan et al. (2018) and Poliak et al. (2018) show that classifiers correctly classified the hypotheses alone in about 69% of SNLI corpus. They further discover that crowd workers adopted specific annotation strategies and heuristics for data generation. Chen et al. (2016) identify similar issues exist with automatically-constructed benchmarks for question-answering (Hermann et al., 2015). Kaushik & Lipton (2018) discover that reported numbers in question-answering benchmarks could often be achieved by the same models when restricted to be blind either to the question or to the passages. Dixon et al. (2018); Zhao et al. (2018) and Kiritchenko & Mohammad (2018) showed how imbalances in training data lead to unintended bias in the resulting models, and, consequently, potentially unfair applications. Shen et al. (2018) substitute words to test the behavior of sentiment analysis algorithms in the presence of stylistic variation, finding that similar word pairs produce significant differences in sentiment score.

Several papers explore richer feedback mechanisms for classification. Some ask annotators to highlight *rationales*, spans of text indicative of the label (Zaidan et al., 2007; Zaidan & Eisner, 2008; Poulis & Dasgupta, 2017). For each document, Zaidan et al. remove the *rationales* to generate *contrast* documents, learning classifiers to distinguish original documents from their *contrasting* counterparts. While this feedback is easier to collect than ours, how to leverage it for training deep NLP models, where features are not neatly separated, remains less clear.

Lu et al. (2018) programmatically alter text to invert gender bias and combined the original and manipulated data yielding gender-balanced dataset for learning word embeddings. In the simplest experiments, they swap each gendered word for its other-gendered counterpart. For example, *the doctor ran because he is late* becomes *the doctor ran because she is late*. However, they do not substitute names even if they co-refer to a gendered pronoun. Building on their work, Zmigrod et al. (2019) describe a data augmentation approach for mitigating gender stereotypes associated with animate nouns for morphologically-rich languages like Spanish and Hebrew. They use a Markov random field to infer how the sentence must be modified while altering the grammatical gender of particular nouns to preserve morpho-syntactic agreement. In contrast, Maudslay et al. (2019) describe a method for probabilistic automatic in-place substitution of gendered words in a corpus. Unlike Lu et al., they propose an explicit treatment of first names by pre-defining name-pairs for swapping, thus expanding Lu et al.'s list of gendered word pairs significantly.

## 3 DATA COLLECTION

We use Amazon's Mechanical Turk crowdsourcing platform to recruit editors to revise each document. To ensure high quality of the collected data, we restricted the pool to U.S. residents that had already completed at least 500 HITs and had an over 97% HIT approval rate. For each HIT, we conducted pilot tests to identify appropriate compensation per assignment, receive feedback from workers and revise our instructions accordingly. A total of 713 workers contributed throughout the whole process, of which 518 contributed edits reflected in the final datasets.

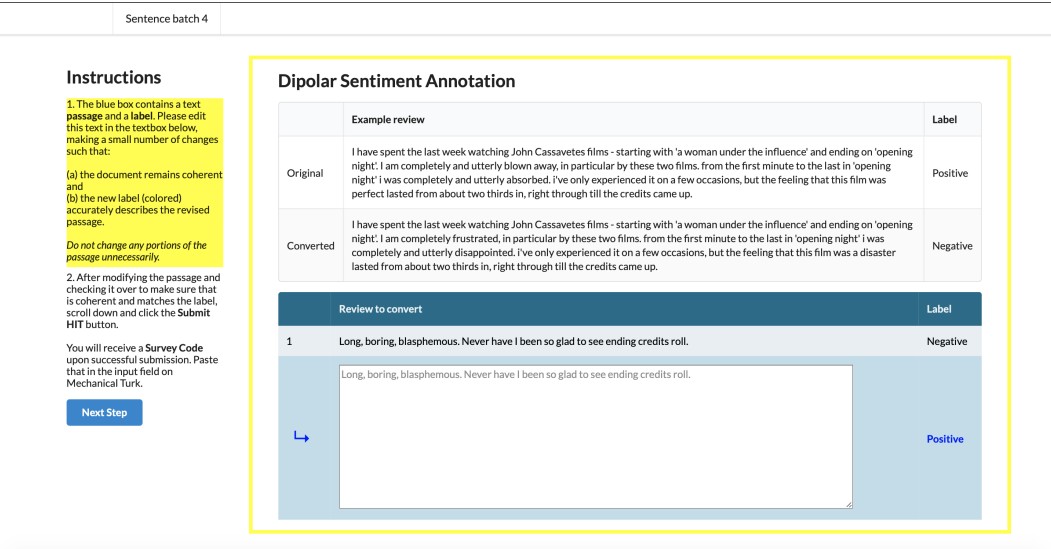

Figure 2: Annotation platform for collecting counterfactually annotated data for sentiment analysis

Table 1: Percentage of inter-editor agreement for counterfactually-revised movie reviews

| Type | Number of tokens | | | | | | | |
|---|---|---|---|---|---|---|---|---|
| | 0-50 | 51-100 | 101-150 | 151-200 | 201-250 | 251-300 | 301-329 | Full |
| Replacement | 35.6 | 25.7 | 20.0 | 17.2 | 15.0 | 14.8 | 11.6 | 19.3 |
| Insertion | 27.7 | 20.8 | 14.4 | 12.2 | 11.0 | 11.5 | 07.6 | 14.3 |
| Combined | 41.6 | 32.7 | 26.3 | 23.4 | 21.6 | 20.3 | 16.2 | 25.5 |

**Sentiment Analysis** The original IMDb dataset consists of $50k$ reviews divided equally across train and test splits. To keep the task of editing from growing unwieldy, we filter out the longest $20\%$ of reviews, leaving $20k$ reviews in the train split from which we randomly sample $2.5k$ reviews, enforcing a 50:50 class balance. Following revision by the crowd workers, we partition this dataset into train/validation/test splits containing $1707$, $245$ and $488$ examples, respectively. We present each review to two workers, instructing them to revise the review such that (a) the counterfactual label applies; (b) the document remains coherent; and (c) no unecessary modifications are made.

Over a four week period, we manually inspected each generated review and rejected the ones that were outright wrong (sentiment was still the same or the review was a spam). After review, we rejected roughly $2\%$ of revised reviews. For 60 original reviews, we did not approve any among the counterfactually-revised counterparts supplied by the workers. To construct the new dataset, we chose one revised review (at random) corresponding to each original review. In qualitative analysis, we identified eight common patterns among the edits (Table 2).

By comparing original reviews to their counterfactually-revised counterparts we gain insight into which aspects are causally relevant. To analyze inter-editor agreement, we mark indices corresponding to replacements and insertions, representing the edits in each original review by a binary vector. Using these representations, we compute the Jaccard similarity between the two reviews (Table 1), finding it to be negatively correlated with the length of the review.

**Natural Language Inference** Unlike sentiment analysis, SNLI is 3-way classification task, with inputs consisting of two sentences, a *premise* and a *hypothesis* and the three possible labels being *entailment*, *contradiction*, and *neutral*. The label is meant to describe the relationship between the facts stated in each sentence. We randomly sampled $1750$, $250$, and $500$ pairs from the train, validation, and test sets of SNLI respectively, constraining the new data to have balanced classes. In one HIT, we asked workers to revise the hypothesis while keeping the premise intact, seeking edits corresponding to each of the two counterfactual classes. We refer to this data as Revised Hypothesis

Table 2: Most prominent categories of edits performed by humans for sentiment analysis (Original/Revised, in order). Red spans were replaced by Blue spans.

| Types of Revisions | Examples |
| --- | --- |
| Recasting *fact* as *hoped for* | The world of Atlantis, hidden beneath the earth's core, is fantastic
The world of Atlantis, hidden beneath the earth's core is **supposed** to be fantastic |
| Suggesting sarcasm | thoroughly captivating **thriller-drama, taking a deep and realistic** view
thoroughly mind numbing **"thriller-drama", taking a "deep" and "realistic" (who are they kidding?)** view |
| Inserting modifiers | The presentation of simply Atlantis' landscape and setting
The presentation of Atlantis' **predictable** landscape and setting |
| Replacing modifiers | "Election" is a highly fascinating and thoroughly **captivating** thriller-drama
"Election" is a highly expected and thoroughly **mind numbing** "thriller-drama" |
| Inserting phrases | Although there's hardly any action, the ending is still shocking.
Although there's hardly any action **(or reason to continue watching past 10 minutes)**, the ending is still shocking. |
| Diminishing via qualifiers | which, while usually containing some reminder of harshness, become **more and more intriguing**.
which, usually containing some reminder of harshness, became **only slightly more intriguing**. |
| Differing perspectives | Granted, **not all of the story makes full sense**, but the film doesn't feature any amazing new computer-generated visual effects.
Granted, **some of the story makes sense**, but the film doesn't feature any amazing new computer-generated visual effects. |
| Changing ratings | one of the worst ever scenes in a sports movie. **3 stars out of 10**.
one of the wildest ever scenes in a sports movie. **8 stars out of 10**. |

(RH). In another HIT, we asked workers to revise the original premise, while leaving the original hypothesis intact, seeking similar edits, calling it Revised Premise (RP).

Following data collection, we employed a different set of workers to verify whether the given label accurately described the relationship between each premise-hypothesis pair. We presented each pair to three workers and performed a majority vote. When all three reviewers were in agreement, we approved or rejected the pair based on their decision, else, we verified the data ourselves. Finally, we only kept premise-hypothesis pairs for which we had valid revised data in both RP and RH, corresponding to both counterfactual labels. As a result, we discarded $\approx 9\%$ data. RP and RH, each comprised of 3332 pairs in train, 400 in validation, and 800 in test, leading to a total of 6664 pairs in train, 800 in validation, and 1600 in test in the revised dataset. In qualitative analysis, we identified some common patterns among hypothesis and premise edits (Table 3, 4).

We collected all data after IRB approval and measured the time taken to complete each HIT to ensure that all workers were paid more than the federal minimum wage. During our pilot studies, workers spent roughly 5 minutes per revised review, and 4 minutes per revised sentence (for NLI). We paid workers $0.65 per revision, and $0.15 per verification, totalling $10778.14 for the study.

## 4 MODELS

Our experiments rely on the following five models: Support Vector Machines (SVMs), Naïve Bayes (NB) classifiers, Bidirectional Long Short-Term Memory Networks (Bi-LSTMs; Graves & Schmidhuber, 2005), ELMo models with LSTM, and fine-tuned BERT models (Devlin et al., 2019). For brevity, we discuss only implementation details necessary for reproducibility.

Table 3: Analysis of edits performed by humans for NLI hypotheses. P denotes *Premise*, OH denotes *Original Hypothesis*, and NH denotes *New Hypothesis*.

| Types of Revisions | Examples |
| --- | --- |
| Modifying/removing actions | **P:** A young dark-haired woman crouches on the banks of a river while washing dishes.
**OH:** A woman washes dishes in the river **while camping**. (Neutral)
**NH:** A woman washes dishes in the river. (Entailment) |
| Substituting entities | **P:** Students are inside of a lecture hall.
**OH:** Students are **indoors**. (Entailment)
**NH:** Students are **on the soccer field**. (Contradiction) |
| Adding details to entities | **P:** An older man with glasses raises his eyebrows in surprise.
**OH:** The man **has no glasses**. (Contradiction)
**NH:** The man **wears bifocals**. (Neutral) |
| Inserting relationships | **P:** A blond woman speaking to a brunette woman with her arms crossed.
**OH:** A woman is talking to **another woman**. (Entailment)
**NH:** A woman is talking to **a family member**. (Neutral) |
| Numerical modifications | **P:** Several farmers bent over working on the fields while lady with a baby and four other children accompany them.
**OH:** The lady has **three** children. (Contradiction)
**NH:** The lady has **many** children. (Entailment) |
| Using/Removing negation | **P:** An older man with glasses raises his eyebrows in surprise.
**OH:** The man **has no** glasses. (Contradiction)
**NH:** The man **wears** glasses. (Entailment) |
| Unrelated hypothesis | **P:** A female athlete in crimson top and dark blue shorts is running on the street.
**OH:** A woman is **sitting on** a white couch. (Contradiction)
**NH:** A woman **owns** a white couch. (Neutral) |

**Standard Methods**  We use `scikit-learn` (Pedregosa et al., 2011) implementations of SVMs and Naïve Bayes for sentiment analysis. We train these models on TF-IDF bag of words feature representations of the reviews. We identify parameters for both classifiers using grid search conducted over the validation set.

**Bi-LSTM**  When training Bi-LSTMs for sentiment analysis, we restrict the vocabulary to the most frequent $20k$ tokens, replacing out-of-vocabulary tokens by `UNK`. We fix the maximum input length at 300 tokens and pad smaller reviews. Each token is represented by a randomly-initialized 50-dimensional embedding. Our model consists of a bidirectional LSTM (hidden dimension 50) with recurrent dropout (probability 0.5) and global max-pooling following the embedding layer. To generate output, we feed this (fixed-length) representation through a fully-connected hidden layer with ReLU (Nair & Hinton, 2010) activation (hidden dimension 50), and then a fully-connected output layer with softmax activation. We train all models for a maximum of 20 epochs using Adam (Kingma & Ba, 2015), with a learning rate of $1e-3$ and a batch size of 32. We apply early stopping when validation loss does not decrease for 5 epochs. We also experimented with a larger Bi-LSTM which led to overfitting. We use the architecture due to Poliak et al. (2018) to evaluate hypothesis-only baselines.[2]

**ELMo-LSTM**  We compute contextualized word representations (ELMo) using character-based word representations and bidirectional LSTMs (Peters et al., 2018). The module outputs a 1024-dimensional weighted sum of representations from the 3 Bi-LSTM layers used in ELMo. We represent each word by a 128-dimensional embedding concatenated to the resulting 1024-dimensional ELMo representation, leading to a 1152-dimensional hidden representation. Following Batch Normalization, this is passed through an LSTM (hidden size 128) with recurrent dropout (probability

---

[2]https://github.com/azpoliak/hypothesis-only-NLI

Table 4: Analysis of edits performed by humans for NLI premises. OP denotes *Original Premise*, NP denotes *New Premise*, and H denotes *Hypothesis*.

| Types of Revisions | Examples |
|---|---|
| Introducing direct evidence | **OP:** Man walking with tall buildings with reflections behind him. (Neutral)
**NP:** Man walking **away from his friend**, with tall buildings with reflections behind him. (Contradiction)
**H:** The man was walking to meet a friend. |
| Introducing indirect evidence | **OP:** An Indian man standing on the bank of a river. (Neutral)
**NP:** An Indian man standing **with only a camera** on the bank of a river. (Contradiction)
**H:** He is fishing. |
| Substituting entities | **OP:** A young man in front of a **grill** laughs while pointing at something to his left. (Entailment)
**NP:** A young man in front of a **chair** laughs while pointing at something to his left. (Neutral)
**H:** A man is outside |
| Numerical modifications | **OP:** The exhaustion in the woman's face while she continues to ride her bicycle in the competition. (Neutral)
**NP:** The exhaustion in the woman's face while she continues to ride her bicycle in the competition **for people above 7 ft**. (Entailment)
**H:** A tall person on a bike |
| Reducing evidence | **OP:** The girl in yellow shorts and white jacket has a tennis ball **in her left pocket**. (Entailment)
**NP:** The girl in yellow shorts and white jacket has a tennis ball. (Neutral)
**H:** A girl with a tennis ball in her pocket. |
| Using abstractions | **OP:** An elderly **woman** in a crowd pushing a wheelchair. (Entailment)
**NP:** An elderly **person** in a crowd pushing a wheelchair. (Neutral)
**H:** There is an elderly woman in a crowd. |
| Substituting evidence | **OP:** A woman is **cutting something with scissors**. (Entailment)
**NP:** A woman is **reading something about scissors**. (Contradiction)
**H:** A woman uses a tool |

0.2). The output from this LSTM is then passed to a fully-connected output layer with softmax activation. We train this model for up to 20 epochs with same early stopping criteria as for Bi-LSTM, using the Adam optimizer with a learning rate of $1e-3$ and a batch size of 32.

**BERT**    We use an off-the-shelf uncased BERT Base model, fine-tuning for each task.[3] To account for BERT's sub-word tokenization, we set the maximum token length is set at 350 for sentiment analysis and 50 for NLI. We fine-tune BERT up to 20 epochs with same early stopping criteria as for Bi-LSTM, using the BERT Adam optimizer with a batch size of 16 (to fit on a Tesla V-100 GPU). We found learning rates of $5e-5$ and $1e-5$ to work best for sentiment analysis and NLI respectively.

---

[3]https://github.com/huggingface/pytorch-transformers

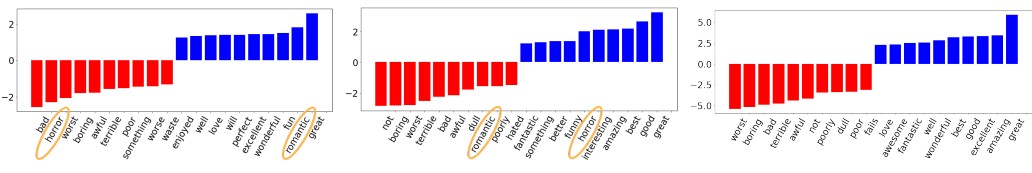

(a) Trained on the original dataset    (b) Trained on the revised dataset    (c) Trained on combined dataset

Figure 3: Most important features learned by an SVM classifier trained on TF-IDF bag of words.

## 5 EXPERIMENTAL RESULTS

**Sentiment Analysis** We find that for sentiment analysis, linear models trained on the original $1.7k$ reviews achieve $80\%$ accuracy when evaluated on original reviews but only $51\%$ (level of random guessing) on revised reviews (Table 5). Linear models trained on revised reviews achieve $91\%$ accuracy on revised reviews but only $58.3\%$ on the original test set. We see similar pattern for Bi-LSTMs where accuracy drops substantially in both directions. Interestingly, while BERT models suffer drops too, they are less pronounced, perhaps a benefit of the exposure to a larger dataset where the spurious patterns may not have held. Classifiers trained on combined datasets perform well on both, often within $\approx 3$ pts of models trained on the same amount of data taken only from the original distribution. Thus, there may be a price to pay for breaking the reliance on spurious associations, but it may not be substantial.

We also conduct experiments to evaluate our sentiment models vis-a-vis their generalization out-of-domain to new domains. We evaluate models on Amazon reviews (Ni et al., 2019) on data aggregated over six genres: *beauty, fashion, appliances, giftcards, magazines,* and *software*, the Twitter sentiment dataset (Rosenthal et al., 2017),[4] and Yelp reviews released as part of the Yelp dataset challenge. We show that in almost all cases, models trained on the counterfactually-augmented IMDb dataset perform better than models trained on comparable quantities of original data.

To gain intuition about what is learnable absent the edited spans, we tried training several models on passages where the edited spans have been removed from training set sentences (but not test set). SVM, Naïve Bayes, and Bi-LSTM achieve $57.8\%, 59.1\%, 60.2\%$ accuracy, respectively, on this task. Notably, these passages are predictive of the (true) label despite being semantically compatible with the counterfactual label. However, BERT performs worse than random guessing.

In one simple demonstration of the benefits of our approach, we note that seemingly irrelevant words such as: *romantic, will, my, has, especially, life, works, both, it, its, lives* and *gives* (correlated with positive sentiment), and *horror, own, jesus, cannot, even, instead, minutes, your, effort, script, seems* and *something* (correlated with negative sentiment) are picked up as high-weight features by linear models trained on either original or revised reviews as top predictors. However, because humans never edit these during revision owing to their lack of semantic relevance, combining the original and revised datasets breaks these associations and these terms cease to be predictive of sentiment (Fig 4). Models trained on original data but at the same scale as combined data are able to perform slightly better on the original test set but still fail on the revised reviews. All models trained on $19k$ original reviews receive a slight boost in accuracy on revised data (except Naïve Bayes), yet their performance significantly worse compared to specialized models. Retraining models on a combination of the original $19k$ reviews with revised $1.7k$ reviews leads to significant increases in accuracy for all models on classifying revised reviews, while slightly improving the accuracy on classifying the original reviews. This underscores the importance of including counterfactually-revised examples in training data.

**Natural Language Inference** Fine-tuned on $1.67k$ original sentence pairs, BERT achieves $72.2\%$ accuracy on SNLI dataset but it is only able to accurately classify $39.7\%$ sentence pairs from the RP set (Table 7). Fine-tuning BERT on the full SNLI training set ($500k$ sentence pairs) results in similar behavior. Fine-tuning it on RP sentence pairs improves its accuracy to $66.3\%$ on RP but causes a drop of roughly 20 pts on SNLI. On RH sentence pairs, this results in an accuracy of $67\%$ on RH and $71.9\%$ on SNLI test set but $47.4\%$ on the RP set. To put these numbers in context, each

---

[4]We use the development set as test data is not public.

Table 5: Accuracy of various models for sentiment analysis trained with various datasets. Orig. denotes *original*, *Rev.* denotes revised, and *Orig. - Edited* denotes the original dataset where the edited spans have been removed.

| Training data | SVM | | NB | | ELMo | | Bi-LSTM | | BERT | |
|---|---|---|---|---|---|---|---|---|---|---|
| | O | R | O | R | O | R | O | R | O | R |
| Orig. (1.7$k$) | **80.0** | 51.0 | **74.9** | 47.3 | **81.9** | 66.7 | **79.3** | 55.7 | **87.4** | 82.2 |
| Rev. (1.7$k$) | 58.3 | **91.2** | 50.9 | **88.7** | 63.8 | **82.0** | 62.5 | **89.1** | 80.4 | **90.8** |
| Orig. − Edited | 57.8 | − | 59.1 | − | 50.3 | − | 60.2 | − | 49.2 | − |
| Orig. & Rev. (3.4$k$) | 83.7 | **87.3** | 86.1 | **91.2** | 85.0 | **92.0** | 81.5 | **92.0** | 88.5 | **95.1** |
| Orig. (3.4$k$) | **85.1** | 54.3 | 82.4 | 48.2 | 82.4 | 61.1 | 80.4 | 59.6 | **90.2** | 86.1 |
| Orig. (19$k$) | **87.8** | 60.9 | 84.3 | 42.8 | 86.5 | 64.3 | 86.3 | 68.0 | 93.2 | 88.3 |
| Orig. (19$k$) & Rev. | **87.8** | 76.2 | 85.2 | 48.4 | 88.3 | 84.6 | 88.7 | 79.5 | 93.2 | 93.9 |

Table 6: Accuracy of various sentiment analysis models on out-of-domain data

| Training data | SVM | NB | ELMo | Bi-LSTM | BERT |
|---|---|---|---|---|---|
| Accuracy on Amazon Reviews | | | | | |
| Orig. & Rev. (3.4$k$) | **77.1** | **82.6** | 78.4 | **82.7** | **85.1** |
| Orig. (3.4$k$) | 74.7 | 66.9 | **79.1** | 65.9 | 80.0 |
| Accuracy on Semeval 2017 (Twitter) | | | | | |
| Orig. & Rev. (3.4$k$) | **66.5** | **73.9** | 70.0 | **68.7** | **82.9** |
| Orig. (3.4$k$) | 61.2 | 64.6 | **69.5** | 55.3 | 79.3 |
| Accuracy on Yelp Reviews | | | | | |
| Orig. & Rev. (3.4$k$) | **87.6** | **89.6** | **87.2** | **86.2** | **89.4** |
| Orig. (3.4$k$) | 81.8 | 77.5 | 82.0 | 78.0 | 85.3 |

Table 7: Accuracy of BERT on NLI with various train and eval sets.

| Train/Eval | Original | RP | RH | RP & RH |
|---|---|---|---|---|
| Original (1.67$k$) | 72.2 | 39.7 | 59.5 | 49.6 |
| Revised Premise (RP; 3.3$k$) | 50.6 | 66.3 | 50.1 | 58.2 |
| Revised Hypothesis (RH; 3.3$k$) | 71.9 | 47.4 | 67.0 | 57.2 |
| RP & RH (6.6$k$) | 64.7 | 64.6 | 67.8 | 66.2 |
| Original w/ RP & RH (8.3$k$) | 73.5 | **64.6** | **69.6** | **67.1** |
| Original (8.3$k$) | **77.8** | 44.6 | 66.1 | 55.4 |
| Original (500$k$) | 90.4 | 54.3 | 74.3 | 64.3 |

individual hypothesis sentence in RP is associated with two labels, each in the presence of a different premise. A model that relies on hypotheses only would at best perform slightly better than choosing the majority class when evaluated on this dataset. However, fine-tuning BERT on a combination of RP and RH leads to consistent performance on all datasets as the dataset design forces models to look at both premise and hypothesis. Combining original sentences with RP and RH improves these

Table 8: Accuracy of Bi-LSTM classifier trained on hypotheses only

| Train/Test | Original | RP | RH | RP & RH |
|---|---|---|---|---|
| Majority class | 34.7 | 34.6 | 34.6 | 34.6 |
| RP & RH ($6.6k$) | **32.4** | **35.1** | **33.4** | **34.2** |
| Original w/ RP & RH ($8.3k$) | 44.0 | 25.8 | 43.2 | **34.5** |
| Original ($8.3k$) | 60.2 | 20.5 | 46.6 | **33.6** |
| Original ($500k$) | 69.0 | 15.4 | 53.2 | **34.3** |

Table 9: Accuracy of models trained to differentiate between original and revised data

| Model | IMDb | SNLI/RP | SNLI/RH |
|---|---|---|---|
| Majority class | 50.0 | 66.7 | 66.7 |
| SVM | 67.4 | 46.6 | 51.0 |
| NB | 69.2 | **66.7** | 66.6 |
| BERT | **77.3** | 64.8 | **69.7** |

numbers even further. We compare this with the performance obtained by fine-tuning it on $8.3k$ sentence pairs sampled from SNLI training set, and show that while the two perform roughly within 4 pts of each other when evaluated on SNLI, the former outperforms latter on both RP and RH.

To further isolate this effect, Bi-LSTM trained on SNLI hypotheses only achieves $69\%$ accuracy on SNLI test set, which drops to $44\%$ if it is retrained on combination of original, RP and RH data (Table 8). Note that this combined dataset consists of five variants of each original premise-hypothesis pair. Of these five pairs, three consist of the same hypothesis sentence, each associated with different truth value given the respective premise. Using these hypotheses only would provide conflicting feedback to a classifier during training, thus causing the drop in performance. Further, we notice that the gain of the latter over majority class baseline comes primarily from the original data, as the same model retrained only on RP and RH data experiences a further drop of $11.6\%$ in accuracy, performing worse than just choosing the majority class at all times.

One reasonable concern might be that our models would simply distinguish whether an example were from the original or revised dataset and thereafter treat them differently. The fear might be that our models would exhibit a hypersensitivity (rather than insensitivity) to domain. To test the potential for this behavior, we train several models to distinguish between original and revised data (Table 9). BERT identifies original reviews from revised reviews with $77.3\%$ accuracy. In case of NLI, BERT and Naïve Bayes perform roughly within 3 pts of the majority class baseline ($66.7\%$) whereas SVM performs substantially worse.

## 6 CONCLUSION

By leveraging humans not only to provide labels but also to intervene upon the data, revising documents to accord with various labels, we can elucidate the difference that makes a difference. Moreover, we can leverage the augmented data to train classifiers less dependent on spurious associations. Our study demonstrates the promise of leveraging human-in-the-loop feedback to disentangle the spurious and non-spurious associations, yielding classifiers that hold up better when spurious associations do not transport out of domain. Our methods appear useful on both sentiment analysis and NLI, two contrasting tasks. In sentiment analysis, expressions of opinion matter more than stated facts, while in NLI this is reversed. SNLI poses another challenge in that it is a 3-class classification task using two input sentences. In future work, we will extend these techniques, leveraging humans in the loop to build more robust systems for question answering and summarization.

## ACKNOWLEDGEMENTS

The authors are grateful to Amazon AWS and NVIDIA for providing GPUs to conduct the experiments, Salesforce Research and Facebook AI for their generous grants that made the data collection possible, Sina Fazelpour, Sivaraman Balakrishnan, Shruti Rijhwani, Shruti Palaskar, Aishwarya Kamath, Michael Collins, Rajesh Ranganath and Sanjoy Dasgupta for their valuable feedback, and Tzu-Hsiang Lin for his generous help in creating the data collection platform. We also thank Abridge AI, UPMC, the Center for Machine Learning in Health, and the AI Ethics and Governance Fund for their support of our broader research on robust machine learning.

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

APPENDIX

Table 10: Most frequent insertions/deletions by human annotators for sentiment analysis.

| Revision | Removed words | Inserted words |
|---|---|---|
| Positive to Negative | *movie, film, great, like, good, really, would, see, story, love* | *movie, film, one, like, bad, would, really, even, story, see* |
| Negative to Positive | *bad, even, worst, waste, nothing, never, much, would, like, little* | *great, good, best, even, well, amazing, much, many, watch, better* |

Table 11: Most frequent insertions/deletions by human annotators for SNLI.

| Revision | Removed words | Inserted words |
|---|---|---|
| **Revising Premise** | | |
| Entailment to Neutral | *woman, walking, man, blue, sitting, men, girl, standing, looking, running* | *person, near, child, something, together, people, tall, vehicle, wall, holding* |
| Neutral to Entailment | *man, street, black, water, little, front, young, playing, woman, two* | *waiting, couple, playing, running, getting, making, tall, game, black, happily* |
| Entailment to Contradiction | *blue, people, standing, girl, front, street, red, young, sitting, band* | *sitting, standing, inside, young, women, child, red, men, sits, one* |
| Contradiction to Entailment | *sitting, man, walking, black, blue, people, red, standing, white, street* | *man, sitting, sleeping, woman, sits, eating, playing, park, two, standing* |
| Neutral to Contradiction | *man, woman, people, boy, black, red, standing, young, two, water* | *man, woman, boy, men, alone, sitting, girl, dog, three, one* |
| Contradiction to Neutral | *man, sitting, black, blue, walking, red, standing, street, white, street* | *man, sitting, woman, people, person, near, something, something, sits, black* |
| **Revising Hypothesis** | | |
| Entailment to Neutral | *man, wearing, white, blue, black, shirt, one, young, people, woman* | *people, there, playing, man, person, wearing, outside, two, old, near* |
| Neutral to Entailment | *white, wearing, shirt, black, blue, man, two, standing, young, red* | *playing, wearing, man, two, there, woman, people, men, near, person* |
| Entailment to Contradiction | *man, wearing, white, blue, black, two, shirt, one, young, people* | *people, man, woman, playing, no, inside, person, two, wearing, women* |
| Contradiction to Entailment | *wearing, blue, black, man, white, two, red, shirt, young, one* | *people, there, man, two, wearing, playing, people, men, woman, outside* |
| Neutral to Contradiction | *white, man, wearing, shirt, black, blue, two, standing, woman, red* | *woman, man, there, playing, two, wearing, one, men, girl, no* |
| Contradiction to Neutral | *wearing, blue, black, man, white, two, red, sitting, young, standing* | *people, playing, man, woman, two, wearing, near, tall, men, old* |

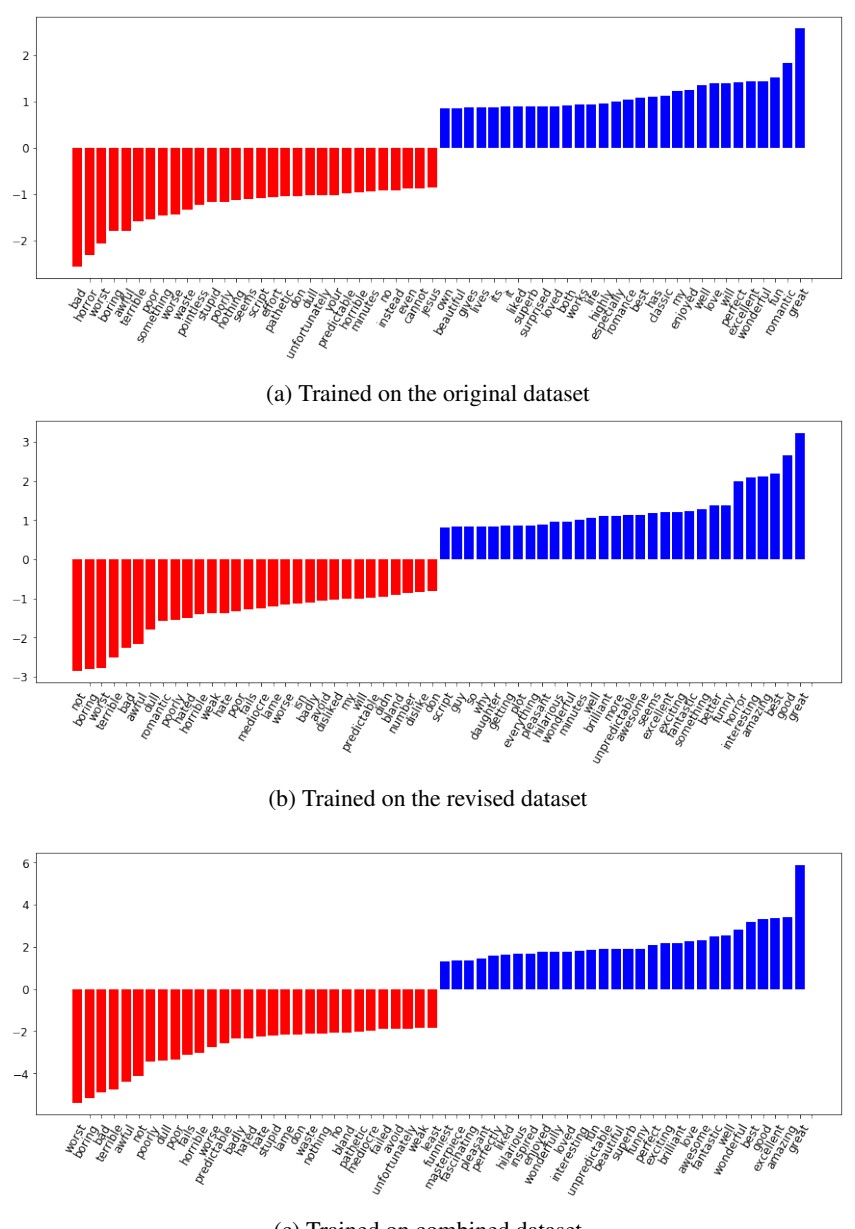

(a) Trained on the original dataset

(b) Trained on the revised dataset

(c) Trained on combined dataset

Figure 4: Thirty most important features learned by an SVM classifier trained on TF-IDF bag of words.

The blue box contains a text passage and a label. Please edit this text in the textbox below, making a small number of changes such that:

(a) the document remains coherent and
(b) the new label (colored) accurately describes the revised passage.

Do not change any portions of the passage unnecessarily.
After modifying the passage and checking it over to make sure that is coherent and matches the label.

(a) Revising IMDb movie reviews

---

The upper blue box contains Sentence 1. The lower blue box contains Sentence 2.
Given that Sentence 1 is True, Sentence 2 (by implication), must either be
(a) definitely True, (b) definitely False, or (c) May be True.

You are presented with an initial Sentence 1 and Sentence 2 and the correct initial relationship label (True, False, or May be True).

Please edit Sentence 2 in the textboxes, making a small number of changes such that:

(a) The new sentences are coherent and
(b) The target labels (in red) accurately describe the truthfulness of the modified Sentence 2 given the original Sentence 1.

Do not change any portions of the sentence unnecessarily.
After modifying the text and checking it over to make sure that it is coherent and matches the target label.

(b) Revising hypothesis in SNLI

---

The upper blue box contains Sentence 1. The lower blue box contains Sentence 2.
Given that Sentence 1 is True, Sentence 2 (by implication), must either be
(a) definitely True, (b) definitely False, or (c) May be True.

You are presented with an initial Sentence 1 and Sentence 2 and the correct initial relationship label (True, False, or May be True).

Please edit Sentence 1 in the textboxes, making a small number of changes such that:

(a) The new sentences are coherent and
(b) The target labels (in red) accurately describe the truthfulness of the original Sentence 2 given the modified Sentence 1.

Do not change any portions of the sentence unnecessarily.
After modifying the text and checking it over to make sure that it is coherent and matches the target label.

(c) Revising premise in SNLI

Figure 5: Instructions used on Amazon Mechanical Turk for data collection

