# OpenReview forum: "Learning The Difference That Makes A Difference With Counterfactually-Augmented Data"
_ICLR.cc/2020/Conference — Accept (Spotlight)_

### Official Review · AnonReviewer3 · 2019-10-24
**Official Blind Review #3**

**Rating:** 8

**Review:**

This paper seeks to separate "causal" features from ones with spurious correlations in the context of natural language machine learning tasks. The proposed approach is to ask human annotators to alter examples in a minimal way that changes the label. Thereby the humans separate out the causal features (those changed) from the spurious or irrelevant features (those left unchanged).

Experiments show that classifiers trained on the original data perform poorly on the altered data and vice versa, but (unsurprisingly) training on the union of the two datasets results in a classifier that performs well in both cases. Furthermore, training an SVM on the original results in irrelevant attributes (such as movie genre) being weighted, whereas these weights are largely removed when training on the union of the datasets. This suggests that the augmented training data results in weighting the "right" features more.

Overall, I think this paper should be accepted because it makes several interesting contributions: It proposes an interesting approach, shows intriguing experimental results, and produces an interesting dataset (size ~2k) that may be useful for future testing.

The main limitation of the paper is that the evidence is largely circumstantial. The method has intuitive appeal and the experimental results are suggestive, but the experiments do not conclusively show that the method achieves something that ordinary machine learning does not.

My suggestion for a further experiment would be to apply the movie review classifiers to, say, book reviews -- something where the task is fundamentally the same but the context is different. If the classifier trained on the union of the original and altered datasets performs better than a classifier trained on only on dataset, then that is strong evidence that this approach yields better extrapolation.


**Experience Assessment:**

I have read many papers in this area.

**Review Assessment: Checking Correctness Of Derivations And Theory:**

N/A

**Review Assessment: Checking Correctness Of Experiments:**

I assessed the sensibility of the experiments.

**Review Assessment: Thoroughness In Paper Reading:**

I read the paper at least twice and used my best judgement in assessing the paper.

---

> ### Author Response · Authors · 2019-11-14
> **Reply to Reviewer 3**
>
> We thank the reviewer for positive feedback and for championing our paper. We are also grateful for your constructive suggestions to improve the paper and would like to report on how we have incorporated your feedback. Inspired by your suggestion, we conducted additional experiments on Amazon Reviews, Yelp Reviews, and Semeval (Twitter) datasets, and found that the counterfactually-augmented data resulted in across-the-board gains. These experiments are featured in the updated draft.

---

### Official Review · AnonReviewer2 · 2019-10-25
**Official Blind Review #2**

**Rating:** 1

**Review:**

The authors propose a new way to augment textual datasets for the task of sentiment analysis, in order to help the learning methods to generalize better by concentrating on learning the different that makes a difference. The main idea of the paper is to augment existing datasets with minimally counteractual versions of them, that change the sentiment of the documents. In this way, all spurious factors will naturally cancel out. The authors use the newly created datasets and show that indeed, the retrained algorithms on the augmented datasets generalize much better.

The main contribution of the paper is the introduction of the idea of counterfactual datasets for sentiment analysis.

Overall, I find the idea of the paper quite interesting and I’m excited to use the datasets they have created. However, I think the relative novelty of the paper does not meet ICLR standards, and it’s better suited as a whitepaper attached to an open dataset release.

**Experience Assessment:**

I have published one or two papers in this area.

**Review Assessment: Checking Correctness Of Derivations And Theory:**

I assessed the sensibility of the derivations and theory.

**Review Assessment: Checking Correctness Of Experiments:**

I assessed the sensibility of the experiments.

**Review Assessment: Thoroughness In Paper Reading:**

I read the paper at least twice and used my best judgement in assessing the paper.

---

> ### Author Response · Authors · 2019-11-14
> **Reply to Reviewer 2**
>
> We thank the reviewer for taking the time to consider our paper and appreciate that you are excited to use our counterfactually-augmented dataset.
>
> While degrees of novelty and the relevant sorts of novelty are a matter of opinion we respectfully assert our view that new ideas, the new resource that we present, and the scientific insights derived from our experiments, are precisely the sorts of novelty that should be sought by conferences.
>
> We respectfully disagree with the reviewer’s suggestion that a fundamentally distinct resource warrants only a whitepaper. We politely point out that many conferences have entire dedicated tracks, and even best paper awards for resources, and that many seminal papers of pivotal importance to the field make precisely this sort of contribution (e.g. ImageNet).
>
> Additionally we point out that the resource is not the only novel idea here. Of chief importance here is the intellectual contribution casting the problem of learning “superficial associations” coherently in the language of intervention, and producing a dataset that addresses counterfactuals in a real sense (as pointed out more eloquently by R1). Moreover, our experiments shed insights about the price to be paid for relying less on spurious associations and our updated experiments (inspired by R3’s suggestions) show that our methods result in improved performance out-of-sample on a variety of datasets.
>
> We hope that you might be willing to reconsider our contributions in light of the significance and uniqueness of the dataset, the insights of our experiments and the demonstrated out-of-domain robustness.

---

### Official Review · AnonReviewer4 · 2019-10-29
**Official Blind Review #4**

**Rating:** 6

**Review:**

This paper addresses the problem of building models for NLP tasks that are robust against spurious correlations in the data by introducing a human-in-the-loop method: annotators are asked to modify data-points minimally in order to change the label.  They refer to this process as counterfactual augmentation.  The authors apply this method to the IMDB sentiment dataset and to SNLI and show (among other things) that many models cannot generalize from the original dataset to the counterfactually-augmented one.

This contribution is timely and addresses a very important problem that needs to be addressed in order to build more robust NLP systems.

Because, however, of a few limitations, I recommend weak acceptance.

My main hesitation comes from a lack of clarity about the main lesson we have learned.  In particular, if the goal is to use this method to augment the data we use to train NLP systems in order to make them more robust, it seems that the time cost of the process will be prohibitive.  On the other hand, perhaps these methods could be used to identify the kind of spurious correlations that models tend to rely on, which could then be used in a more automated data augmentation process.  If that's the goal, however, a more detailed error analysis would need to be included.

A few small comments:

* There was some analysis of the augmented IMDB dataset, but none of the SNLI dataset.  I would love to see a more detailed investigation of what annotators usually did.  For instance, a reason that hypothesis-only models do well is that certain words are very predictive of certain labels (e.g. "not" and contradiction).  Do people leave the negations in when modifying such examples for entailment or neutrality, thus breaking the simple correspondence?  That's a very simple kind of question; more generally, I'd like to see more analysis of the new dataset.

* The BiLSTM they use is very small (embedding and hidden dimension 50).  Given that BERT is most robust against their manipulation, it would be good to see a more powerful recurrent model for comparison.  It would be easy to use ELMo here, if the main question is about Transformers vs recurrent models.


Some very minor / typographic comments:

* abstract: "with revise" should be "with revising"
* first paragraph page 2: some references to causality literature and definition of spuriousness as common cause
* page 2, "We show that..." I'd break this into two sentences to make it easier to parse.
* Table 3: I would make two columns for each model with accuracy on original versus revised.  With the current table, one has to compare cells in the top half of the table to those in the bottom half of the table, which is quite difficult to do.

**Experience Assessment:**

I have published one or two papers in this area.

**Review Assessment: Checking Correctness Of Derivations And Theory:**

N/A

**Review Assessment: Checking Correctness Of Experiments:**

I assessed the sensibility of the experiments.

**Review Assessment: Thoroughness In Paper Reading:**

I made a quick assessment of this paper.

---

> ### Author Response · Authors · 2019-11-14
> **Reply to Reviewer 4**
>
> Thanks for the detailed and thoughtful review. We are glad that you think of this paper as a timely contribution addressing an important problem that must be addressed in order to build more robust NLP systems.
>
> We agree with your point that it would be great to have a practical takeaway guiding practitioners for what to do in practice. We believe that the first step here is to characterize the problem coherently and that having laid this groundwork, one immediate next step is, as you suggest, to develop a more practical solution that requires a less expensive/onerous annotation effort.
>
> The key contribution of our paper is to provide a clear characterization of a variety of concerns in the language of interventions and to demonstrate that indeed, they can be addressed by acquiring interventional data. The knowledge that (i) NLP models trained on counterfactually augmented data suffer less from these problems and (ii) transport better out of sample (see new results in the updated draft, per R3’s suggestion) validates this.
>
> As you mentioned, our solution requires significant expenditure (both financial and human capital) compared to simply labeling data. As a follow-up, for existing datasets, our next steps include investigating how to make these adjustments in a cost-effective way. In preliminary work, we have been investigating how to use humans in the loop more effectively. One approach involves using generative models to propose candidate substitutions and relying on humans only accept or reject the revisions (vs having to write them from scratch). Our experience with crowdsourcing suggests that this feedback would be significantly cheaper to collect (provided that a reasonable fraction of suggestions were appropriate).
>
> We additionally note that for some tasks, such as NLI, creating new datasets already requires annotators to synthesize examples de novo and the fractional increase for soliciting counterfactually-augmented data might not be as onerous as compared to tasks where the default is to rely on annotators only for tags.
>
> We are also appreciative of your constructive suggestions to improve the paper, and have taken several steps to improve the draft. These include updating the draft to include (i) a detailed analysis of edits performed on SNLI, (ii) results on various datasets using an ELMo based classifier; (iii) concerning your question about larger Bi-LSTMs, we had tried a large Bi-LSTM but it overfit badly. We have updated the draft to include this detail.
>
> Thanks also for catching several typographic errors. We have addressed them in the new draft.

---

### Official Review · AnonReviewer1 · 2019-11-07
**Official Blind Review #1**

**Rating:** 8

**Review:**

Summary:
       The authors take two tasks,sentiment analysis and natural language inference, and identify datasets for them which they counterfactually augment it by asking people over the Amazon Mechanical Turk Platform to change either the sentiment (in the case of sentiment analysis) or the nature of relationship in the NLI task by making minimal changes to the text that produce the targeted changes.

Authors find that popular models trained on either fail on the other dataset while the models trained on both actually generalize much better. This is because the original sample and its counterfactual pair the label changed , has the difference in the text that matters to the change and this pair could reduce spurious correlations that models might find in the data distribution.

Pros:
 This is a very interesting experiment and certainly the dataset that will be released would be extremely valuable to the community. The one part (I dont have much NLP background but I do have a causality background) that I like most is that the new text generated are counterfactual in some real sense with respect to a real world generating process - that is people modifying text with changed targets.

 A lot of existing work that claim to do counterfactual changes do not specify assumptions about the generating mechanism. For counterfactuals to be valid they have to be intervention on the actual generating mechanism (or an assumed one) acting on a given unit (latent) that produced the current sample. The paper in that respect (even if it does not explicitly specify relationship between counterfactuals and generating mechanisms) tries to be faithful to a "strict causal notion" by actually asking people to modify the text.

Cons:
    - I think the authors want to make an explicit connection to counterfactuals as understood in the causality community. Then they shy away from it saying they are inspired by it. May be a formal exposition in the supplement about counterfactuals and generating mechanisms could help readers from other communities (NLP) even it means repeating standard/synthetic examples. Its good to say what exactly in a counterfactual generation process, the "people" in amazon turk were substituting.

   -  Is the romantic/ horror flips and their absence the only spurious thing in Figure 4 ?
  -  In figure 6, it appears that BERT is sensitive to the domain - does it mean that it is bad ? - Authors indicate that ideally it must not be so. Because Table 3 results seem to indicate that BERT performs the best in almost all the cases .
 -  Can the authors highlight the best performances in each case in the Tables by a bold face.  It helps easily eye ball the best performing model.


**Experience Assessment:**

I have read many papers in this area.

**Review Assessment: Checking Correctness Of Derivations And Theory:**

N/A

**Review Assessment: Checking Correctness Of Experiments:**

I assessed the sensibility of the experiments.

**Review Assessment: Thoroughness In Paper Reading:**

I read the paper at least twice and used my best judgement in assessing the paper.

---

> ### Author Response · Authors · 2019-11-14
> **Reply to Reviewer 1**
>
> Thank you for the thoughtful review and positive assessment. We are glad to see that you appreciate the genuine flavor of causality in our paper and support our paper’s acceptance.
>
> We agree that a formal exposition introducing an NLP/deep learning audience to the basics of interventions and counterfactuals and expressing a toy DAG to explain the spurious associations between the review sentiment and the manifestation in text of other attributes of the review, including but not limited to the genre, actors, budget, etc. We are actively working on preparing this exposition and while it is not yet in the draft we plan to have it prepared in advance of the camera-ready version.
>
> We thank the reviewer for pointing out that we should have been more thorough in explaining that while genre is a clear example of such a spurious association, it is far from the only one captured in Figure 4. Indeed, many other words, including “will”, “my”, “has”, “especially”, “life”, “works”, “both”, “it”, “its”, “lives”, “gives”, “own”, “jesus”, “cannot”, “even”, “instead”, “minutes”, “your”, “effort”, “script”, “seems”, and “something”, appear to be spuriously associated with sentiment and are captured by the original-only and revised-only classifiers as highly-weighted features., Notably all of these features fall out from the highly-weighted features when our classifier is trained on counterfactually-augmented data.
>
> Regarding the sensitivity of BERT models, Table 9 shows the ability of a model explicitly trained to differentiate between the original and the revised data. This is to shed some insight on how much the two differ (on account of our intervention). Because the two indeed are different, we expect that a model should be able to differentiate them to some degree. We note that a model class’s ability to differentiate between the original and revised data when explicitly trained to do so may not necessarily be correlated with how susceptible that model is to breaking when evaluated out of sample.
>
> We’re grateful for your comments on exposition and will continue to address these points as we improve the draft.

---

### Author Response · Authors · 2019-11-14
**General reply to reviewers**

We would like to thank all four reviewers for thoughtful reviews. We are glad to see that 3 reviewers vote for acceptance and that two champion the paper, with the reviewers recognizing the paper to be “timely”, to address “an important problem”, to contribute an “exciting” resource, and to be “extremely valuable to the community”.

We are also grateful to the reviewers for a number of constructive suggestions. Inspired by their feedback, we have run several additional experiments and added them to the draft. For example, per R3’s suggestions, we evaluated the performance of our classifiers on out-of-sample data, including Yelp Reviews, Amazon Reviews (aggregated over six different genres), and tweets, finding that indeed models trained on counterfactually-augmented data give better performance out of domain. This finding holds across model classes. We have also conducted experiments using ELMo as suggested by R4, further validating the usefulness of counterfactually augmenting data. We have updated the draft to add these experiments and address other concerns of the reviewers.

Below, we reply to each review in more detail in their respective threads.

---

### Decision · Program_Chairs · 2019-12-19

**Decision:**

Accept (Spotlight)

**Comment:**

This paper introduces the idea of a counterfactually augmented dataset, in which each example is paired with a manually constructed example with a different label that makes the minimal possible edit to the original example that makes that label correct. The paper justifies the value of these datasets as an aid in both understanding and building classifiers that are robust to spurious features, and releases two small examples.

On my reading, this paper presents a very substantially new idea that is relevant to a major ongoing debate in the applied machine learning literature: How do we build models that learn some intended behavior, where the primary evidence we have of that behavior comes in the form of datasets with spurious correlations/artifacts.

One reviewer argued for rejection on the grounds that dataset papers are not appropriate for publication at a main conference. I don't find that argument compelling, and I'm also not sure that it's accurate to call this paper primarily a dataset paper. We could not reach a complete consensus after further discussion. The other reviews raised some additional concerns about the paper, but the revised manuscript appears to have address them to the extent possible.